# Variation of the 2D Pattern of Brain Proteins in Mice Infected with *Taenia crassiceps* ORF Strain

**DOI:** 10.3390/ijms25031460

**Published:** 2024-01-25

**Authors:** Mariana Díaz-Zaragoza, Ricardo Hernández-Ávila, Abraham Landa, Pedro Ostoa-Saloma

**Affiliations:** 1Departamento de Ciencias de la Salud, Centro Universitario de los Valles, Universidad de Guadalajara, Carretera Guadalajara-Ameca Km. 45.5, Guadalajara 46600, Mexico; marina.diaz@academicos.udg.mx; 2Departamento de Inmunología, Instituto de Investigaciones Biomédicas, Universidad Nacional Autónoma de México, Ciudad Universitaria, A.P. 70228, Mexico City 04510, Mexico; ricoavila@iibiomedicas.unam.mx; 3Departamento de Microbiología y Parasitología, Facultad de Medicina, Universidad Nacional Autónoma de México, Ciudad Universitaria, A.P. 70228, Mexico City 04510, Mexico; landap@unam.mx

**Keywords:** *Taenia crassiceps*, brain damage, brain proteome, Neuro-Immuno-Endocrine network

## Abstract

Some parasites are known to influence brain proteins or induce changes in the functioning of the nervous system. In this study, our objective is to demonstrate how the two-dimensional gel technique is valuable for detecting differences in protein expression and providing detailed information on changes in the brain proteome during a parasitic infection. Subsequently, we seek to understand how the parasitic infection affects the protein composition in the brain and how this may be related to changes in brain function. By analyzing de novo-expressed proteins at 2, 4, and 8 weeks post-infection compared to the brains of the control mice, we observed that proteins expressed at 2 weeks are primarily associated with neuroprotection or the initial response of the mouse brain to the infection. At 8 weeks, parasitic infection can induce oxidative stress in the brain, potentially activating signaling pathways related to the response to cellular damage. Proteins expressed at 8 weeks exhibit a pattern indicating that, as the host fails to balance the Neuro-Immuno-Endocrine network of the organism, the brain begins to undergo an apoptotic process and consequently experiences brain damage.

## 1. Introduction

It is well established that certain parasites have the potential to occasionally influence brain proteins, triggering immune responses or nervous system function. One of the most well-known examples is Toxoplasma gondii, a parasite infecting mammals including humans, which has been associated with effects on behavior and brain function [1,2,3,4].

*Taenia crassiceps* (*T. crassiceps*) is a cestode parasite that primarily infects rodents and can exert influences on the host’s immune system and brain. The presence of *T. crassiceps* in the mouse’s body can lead to the modulation of the host’s immune system [5]. It has been observed that infection with *T. crassiceps* can induce immune responses in the brains of mice, which may lead to inflammation and changes in brain function [6,7]. Research on *T. crassiceps* infection in mice has investigated its correlation with behavioral changes, indicating that the presence of the parasite might influence mouse behavior. This finding could be pertinent to understanding how the parasite is transmitted to its definitive hosts, which are typically predators [8]. Human infection is believed to occur following the consumption of food or water contaminated with infectious eggs shed in the feces of carnivores. While all recognized cases involving muscles or subcutaneous tissue in humans have been associated with underlying immunosuppression, there are reported instances that do not seem to require a compromised immune system [9].

Two-dimensional gels, also known as 2D gels, are a powerful technique used in protein research to separate and analyze proteins in complex samples. These gels can be employed to identify differences in protein expression between samples and for the discovery of biomarkers, among other purposes. Two-dimensional gels have been employed in neuroproteomic studies to analyze the proteins present in the mouse brain, providing a better understanding of its composition and changes in response to various conditions, such as diseases or treatments. The results of these studies have provided essential information to advance the understanding of processes such as brain development, aging, neurological diseases, and the response to therapeutic treatments [10,11,12]. A limitation, as pointed out by some researchers when conducting analyses using proteomic methods, is that only the most abundant proteins are identified. Proteins that are expressed at low levels are often not detected. Thus, there is a risk that the approach does not account for all potentially relevant proteins. Although this is possible, it is unlikely given that the resolution level of the technique is very high.

Given that infection with *T. crassiceps* affects both the immune system and brain tissue, the observed changes in brain proteins are likely the result of a complex interaction between the direct effects of the parasite and the host’s immune response. The precise characterization of these effects and their relative contribution may require detailed studies in experimental models and advanced techniques, such as proteomics and mass spectrometry, or bioinformatics to identify specific proteins and their changes in response to the infection.

Research on the effects of *T. crassiceps* on the brain and the specific proteins involved in its interactions with the nervous system remains an active area of study. In this work, we aim to demonstrate how the 2D gel technique is valuable for detecting differences in protein expression and provide detailed information on changes in the brain proteome during a parasitic infection, thus subsequently understanding how parasitic infection affects protein composition in the brain and how this may be related to changes in brain function.

## 2. Results

Figure 1 provides a comprehensive overview of the changes in the expression of specific brain proteins in the infected group compared to the non-infected group. Notably, these differences exhibit variation concerning the duration infection. This figure highlights proteins that undergo modification in their expression, with a particular focus on those proteins that appear “de novo”, in relation to the control group, due to infection. In Figure 2, we present proteins expressed at specific times that are not found in the control brain at those times, indicating potential de novo synthesis in cells. At week 2, there are 12 such proteins, 2 proteins at week 4, and 12 proteins at week 8. 

The identification of proteins in Table 1 is based on comparisons between reports of mouse brain proteins with a matching molecular weight and isoelectric point and our experimental results. The identification draws from data representing the most comprehensive proteome coverage for mammalian brains to date, providing a foundation for future quantitative studies in brain proteomics using mouse models. The proteomic approach presented here may have broad applications for the rapid proteomic analysis of various mouse models of human brain diseases.

In Table 2, the proteins from Table 1 are displayed but as part of a functional group. From Table 2, it can be inferred that the identified proteins do not belong to a specific brain process but rather are ubiquitous proteins that support the idea of a generalized degenerative process in the brain rather than one confined to a particular region.

## 3. Discussion 

In the early stages of infection, it is common to observe a systemic TH1 response involving the production of cytokines, such as interferon-gamma (IFN-γ). The TH1 response is commonly associated with cellular immunity and the fight against intracellular infection. During this initial phase, the immune system may attempt to control and limit the spread of the parasite. The emergence of new brain proteins in mice infected with *T. crassiceps* may be the result of a combination of factors, including the direct action of the parasite and the host’s Neuro-Immuno-Endocrine network response. Often, it is challenging to completely separate the direct effects of the parasite from the immune responses triggered by the infection. The parasite *T. crassiceps* can have a direct impact on the mouse’s brain, either through the release of metabolic products, manipulation of the local immune response, or physical interaction with brain cells. This can influence brain proteins and other components of brain tissue [8].

Infection with *T. crassiceps* will also trigger an immune response from the host. This response may involve the activation of immune cells, the release of cytokines, and other inflammatory mediators in the brain. These changes in the brain environment can have a significant effect on the expression and activity of proteins in brain tissue, potentially influencing the expression of proteins related to inflammation and immune response. Cytokines such as gamma interferon (IFN-γ) and interleukin-6 (IL-6) are common activators of inflammatory signaling pathways in the mouse brain in response to parasitic infections [45].

The proteins listed in Table 1 reflect, or their expression is a consequence of, the series of chemical signals that converge during infection. At 2 weeks post-infection, the expressed proteins are associated with the physiology of brain cells, specifically the protection against anoxia, synaptic plasticity, detoxification, combating oxidative stress, addressing depression, neuronal damage, overcoming anxiety, and responding to inflammation [19,20,21,22,23,24,25,26,27,28,29,30,31,32,33].

Intraperitoneal infection in the mouse cysticercosis model quickly shifts to the TH2 type or even a mixed profile of type 1/type 2 cytokines, which is permissive for parasite growth. The TH2 immune response is characterized by the production of cytokines such as interleukin-4 (IL-4), interleukin-5 (IL-5), and interleukin-13 (IL-13), and is associated with humoral immunity. This results in the unrestricted growth of the parasite, which, in experimental cases, can lead to the death of the animal, demonstrating little or no immunological resistance to parasitic growth. 

On the other hand, during the infection, cells of the central nervous system (CNS) have the ability to produce inflammatory mediators such as chemokines, adhesion molecules, and cytokines [46]. These responses can lead to the significant infiltration of various leukocytes, culminating in pathogen-specific adaptive immune responses in the CNS. The direct recognition of microbial molecules by cells in nervous tissue and the subsequent innate immune response appear to be key elements in protecting the CNS [47]. The inflammatory response in the CNS plays a crucial reparative role and involves the participation of various immune cell types (macrophages, mast cells, T and B lymphocytes, dendritic cells) and resident CNS cells (microglia, astrocytes, neurons), as well as adhesion molecules, cytokines, and chemokines, among other protein components. During neuroinflammation, chemotaxis is a significant event in the recruitment of cells into the CNS. 

The recruitment of lymphocytes involves the presence of chemokines and chemokine receptors, expression of adhesion molecules, interaction between lymphocytes and the blood–brain barrier (BBB) endothelium, and ultimately their passage through the BBB to reach the site of inflammation. The metabolic products released by the parasite, such as lipopolysaccharides or glycoproteins, or the cytokines and mediators of the parasite’s Neuroimmunendocrine network can intermingle with the constitutive signals of the brain, generating the regulation of protein expression mediated by cellular communication pathways. Apparently, by the eighth week, this process is uncontrolled and progressing. Under these conditions, the reparative effects of the inflammatory response are overwhelmed and can promote brain damage [46]. By week 8, the expressed proteins are associated with stress, combating oxidative stress, apoptosis, mobility, and learning. They play a role in regulating membranes and neurotransmission, especially at synapses and myelination. There is also the regulation of energetic homeostasis, and the aim is to control the neurodegeneration that begins to manifest itself [34,35,36,37,38,39,40,41,42,43,44,45,46,47,48,49]. It is possible that there may be protein expression differences at the level of small brain regions, and due to the strategy of processing the entire organ, we may not be able to detect them. The reasoning behind processing the entire brain instead of spatially expressing proteins was that, based on the results obtained, a group of proteins specifically associated with a region or a cell nucleus linked to a function could be identified. However, we did not find evidence of that. According to the results, the found proteins do not belong to a specific cerebral process but are ubiquitous proteins that support the idea of a generalized degenerative process in the brain rather than one confined to a particular region. Another possibility is that by processing the entire brain, the expression change occurring in a limited region of the brain may be “diluted”.

## 4. Materials and Methods

The study utilized female BALB/c strain mice which were housed in the animal facilities of the Faculty of Medicine at UNAM under controlled conditions of temperature (22 °C) in a pathogen-free environment, with a relative humidity of 50 to 60%, 12-h light-dark cycles, and free access to food and water. 

Infection with the *Taenia crassiceps* cysticerci ORF strain and two-dimensional gel electrophoresis (2DE) were performed according to [48,49].

Proteomics Analysis

The 2DE gels were digitized using an HP Scanjet-G4050 scanner with a resolution of 300 DPI and analyzed using PDQuest™ 2DE software version 8.0 (Bio-Rad Laboratories, Inc., Hercules, CA, USA) to determine differences in the expression of proteins depending on the cysticercus infection time. Master images were created for each group from their 3 replicates. In other words, a Master image was obtained for protein extracts from each control group (2, 4, and 8 weeks) and from each of the infected mice (2, 4, and 8 weeks). The coordinates of each spot were calculated according to the isoelectric point markers of the 2DE standards (Bio-Rad Laboratories, Inc., Hercules, CA, USA).

Mouse Brain Protein Extraction

Brain tissue was quickly taken from cryopreserved vials and processed under cold conditions to prevent degradation. For each mouse in the group, a portion of the brain was taken, and tissue pooling was performed, from which proteins were extracted. Each brain tissue pool was manually homogenized with a teflon pestle on ice in 600 µL of 2D buffer (8 M Urea, 50 mM DTT, 2% CHAPS, 2% Ampholine pH 3–10 (Bio-Rad) in the presence of a protease inhibitor (Halt™ Protease Inhibitor Cocktail, Thermo Scientific, Waltham, MA, USA). It was sonicated in cold conditions at 15-s intervals, shaken for 2 h at 4 °C, and then centrifuged for 20 min at 12,000 rpm at 4 °C. The protein concentration in the supernatant was measured using the Bradford method.

Bioinformatic approach. 

A literature search was conducted for articles reporting proteomic analyses and protein identification in the BALB/c mouse brain [12,50,51]. An analysis of metabolic pathways was conducted involving the proteins found using the Kyoto Encyclopedia of Gene and Genomes database (https://www.genome.jp/kegg/ (accessed on 13 January 2024)) and the UNIPROT database (https://www.uniprot.org (accessed on 13 January 2024)).

## 5. Conclusions

By analyzing de novo-expressed brain proteins at 2, 4, and 8 weeks post-infection compared to the brains of the control mice, we observed that the proteins expressed at 2 weeks are primarily associated with neuroprotection or the initial response of the mouse brain to infection. By 8 weeks, parasitic infection may induce oxidative stress in the brain, potentially activating signaling pathways related to the response to cellular damage. The proteins expressed at 8 weeks exhibit a pattern indicating that, when unable to balance the organism’s Neuroimmunendocrine network, the brain begins to undergo an apoptotic process, leading to consequential brain damage. This damage is manifested in previously reported behaviors, including sexual activity, aggression, social status, defense response, as well as the impairment of short-term memory. The characterization of the proteins reported in the study is at the level of the isoelectric point and molecular weight. It is evident that a deeper characterization at the sequence level of the proteins and their recognition by antibodies is required, and this is currently underway.

## Figures and Tables

**Figure 1 ijms-25-01460-f001:**
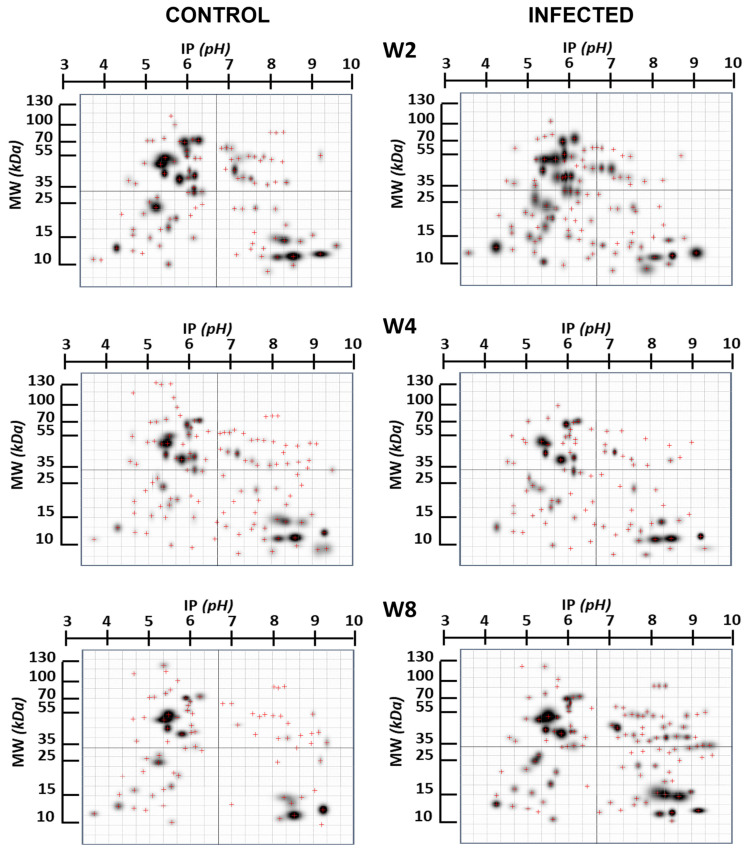
Two-dimensional gels’ image of the brain proteins in infected and control mice at different infection time points. W2, W4, W8: weeks 2, 4, 8, respectively. The red crosses represent the center of the spot.

**Figure 2 ijms-25-01460-f002:**
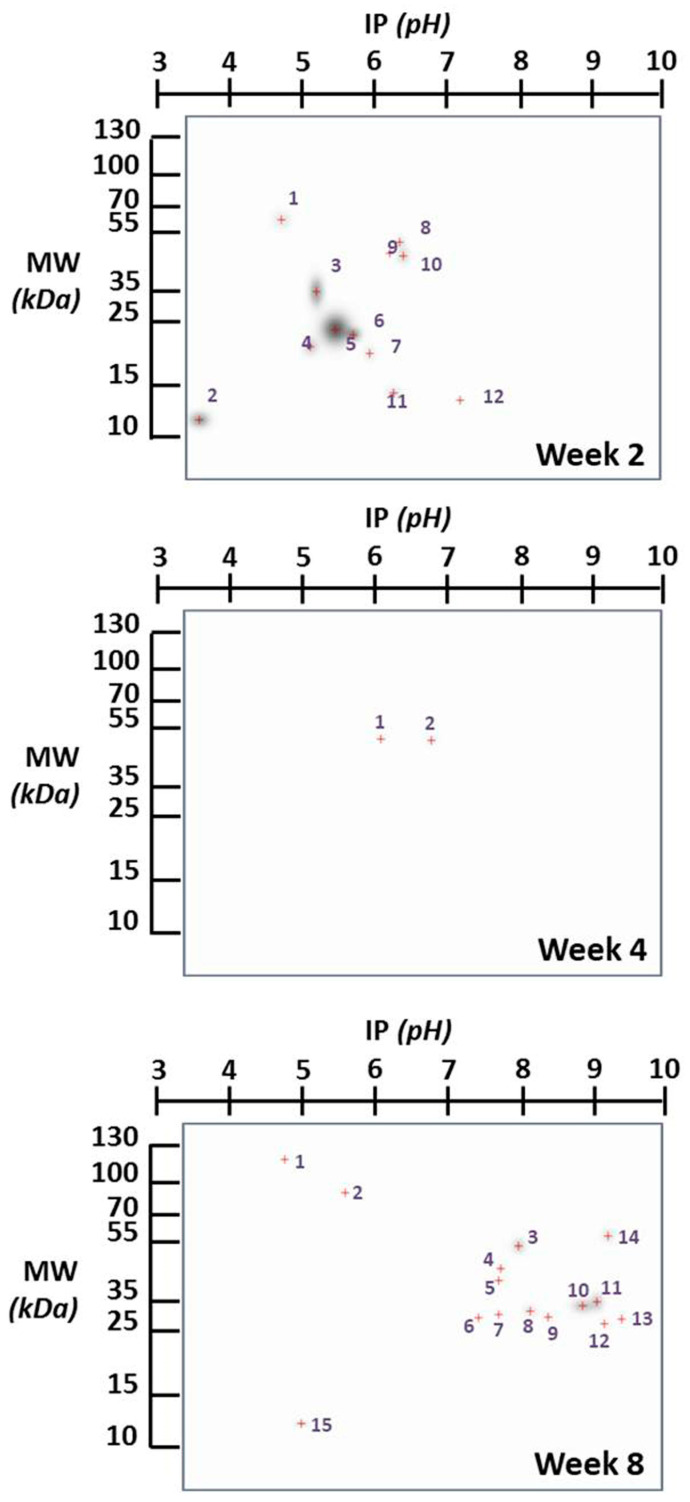
Two-dimensional gels’ image of the brain proteins in infected and control mice at different infection time points. The proteins expressed at the corresponding time, which are not found in the control brain at that time (suggesting the possible de novo synthesis in cells), are exclusively presented. The spots are numbered to identify the proteins in Table 1. The red crosses represent the center of the spot.

**Table 1 ijms-25-01460-t001:** Probable identification of the proteins indicated in Figure 2. The proteins highlighted in black are brain proteins that closely match a high percentage with what is reported in the literature. The proteins highlighted in red are those that approximate a protein reported to one of the values of the experimental spot: either the molecular weight or the isoelectric point.

Infection Time	N° Spot	~MW Spot	~IP Spot	Probable Protein	MW Protein	IP Protein	Refs.
Week 2	1	62.24	4.7	Neurofilament protein NF-66 mRNA α-Internexin	61	4.79	[13]
	2	11.8	3.6	U6 snRNA-associated Sm-like protein LSm3	11.8	4.08	[14]
	3	35	5.2	Creatine kinase b-chain	34.09	5.22	[15]
	4	21.11	5.1	Lactoylglutathione Lyase Phosphatidylethanolamine-binding protein mRNA	21 21	5.15 5.11	[16,17]
	5	23.66	5.45	Glutathione S-transferase P 1	23.6	5.43	[18]
	6	22.78	5.7	γ-Enolase	25.24	5.73	[19]
	7	20	5.95	Diphosphoinositol polyphosphate phosphohydrolase 2 Neuron-specific protein family member 1	20.16 20.93	5.99 5.99	[20]
	8	52	6.35	tRNA modification GTPase GTPBP3, mitochondrial	52.1	6.37	[21]
	9	48	6.2	Basic leucine zipper and W2 domain-containing protein 2	48.06	6.26	[22]
	10	47	6.4	Neuronal pentraxin-2	47.14	6.3	[23]
	11	14.2	6.3	Gelsolin precursor, plasma	17.13	6.21	[24]
	12	13.67	7.2	Serum amyloid A-3 protein	13.7	6.0, 6.4, 7.0, 7.4, 7.5, 8.0	[25]
Infection Time	N° spot	~MW spot	IP spot	Probable protein	MW protein	IP protein	
Week 4	1 2	51.25 50.5	6.05 6.75	Annexin VII (synexin) 4-aminobutyrat aminotransferanse, mitochondrial precursor	50 52.83	6.02 6.78	[26] [27]
Infection Time	N° spot	~MW spot	~IP spot	Probable protein	MW protein	IP protein	
	1	118	4.79	Apg-2 mRNA	120	5.0	[28]
	2	96.67	5.6	Ubiquitin carboxyl-terminal hydrolase	96.7	5.42	[29]
	3	57.75	7.98	*-------------------------------------------------------------------------------*	*-----*	*----*	
	4	45.5	7.75	LanC-like protein 1 Ethanolamine phosphotransferase 1	45.34 45.35	7.57 7.33	[30] [31]
	5	42	7.7	Cytosolic acyl coenzyme A thioester hydrolase mRNA	42.11	7.65	[32]
	6	29.5	7.45	Adaptin ear-binding coat-associated protein 1	29.63	7.72	[33]
Week 8	7	30	7.72	Gap junction gamma-3 protein	30.29	7.71	[34]
	8	31.25	8.14	Thymidine kinase 2, mitochondrial	31.26	8.71	[35]
	9	29.5	8.38	Glyceraldehide 3-phosphate deshydrogenase	28.93	8.39	[36]
	10	33	8.85	Alpha/beta hydrolase domain-containing protein 11	33.56	8.86	[37]
	11	34.5	9.15	AMMECR1-like protein Protein N-terminal asparagine amidohydrolase	34.52 34.59	9.18 9.07	[38] [39]
	12	27	9.18	Mitochondrial fission factor Major prion protein	27.22 27.98	9.1 9.13	[40] [41]
	13	28.75	9.4	Peroxisomal membrane protein 11B	28.72	9.68	[42]
	14	57.07	9.3	Lysophospholipid acyltransferase LPCAT4	57.1	8.65	[43]
	15	12.33	5.0	Eukaryotic translation initiation factor 4E-binding protein 1	12.33	5.32	[44]

**Table 2 ijms-25-01460-t002:** Association of the identified proteins with a functional group within metabolism.

Protein	Pathway	KEEG Code
Neurofilament protein NF-66 mRNA α-Internexin	Cytoskeleton	---
U6 snRNA-associated Sm-like protein LSm3	RNA degradation	mmu:67678
Creatine kinase b-chain	Arginine and proline metabolism	mmu:12709
Lactoylglutathione Lyase	Pyruvate metabolism	mmu:109801
Phosphatidylethanolamine-binding protein mRNA	Peptidase inhibitors	mmu:23980
Glutathione S-transferase P 1	Glutathione metabolism	mmu:14858
γ-Enolase	Glycolysis/Gluconeogenesis	mmu:13807
Diphosphoinositol polyphosphate phosphohydrolase 2	Hydrolase	mmu:71207
Neuron-specific protein family member 1	Membrane trafficking	rno:25247
tRNA modification GTPase GTPBP3, mitochondrial	Transfer Rna biogenesis	mmu:70359
Basic leucine zipper and W2 domain-containing protein 2	Translation regulator	mmu:66912
Neuronal pentraxin-2	Signaling proteins	---
Gelsolin precursor, plasma	Phagocytosis/Regulation of actin cytoskeleton	mmu:56320
Serum amyloid A-3 protein	Exosome	mmu:20210
Annexin VII (synexin)	Amyotrophic lateral sclerosis	mmu:11750
4-aminobutyrat aminotransferanse, mitochondrial precursor	Alanine, aspartate, and glutamate metabolism Valine, leucine, and isoleucine degradation Beta-Alanine metabolism Ropanoate metabolism Butanoate metabolism Metabolic pathways GABAergic synapse	mmu:268860
Apg-2 mRNA	Heat shock protein	---
Ubiquitin carboxyl-terminal hydrolase	DNA repair and recombination proteins Ubiquitin system	mmu:230484
LanC-like protein 1	Glutathione metabolism	mmu:14768
Ethanolamine phosphotransferase 1	Phosphonate and phosphinate metabolism Glycerophospholipid metabolism Ether lipid metabolism	mmu:99712
Cytosolic acyl coenzyme A thioester hydrolase mRNA	Fatty acid elongation Biosynthesis of unsaturated fatty acids Ovarian steroidogenesis	mmu:26897
Adaptin ear-binding coat-associated protein 1	Membrane trafficking	mmu:67602
Gap junction gamma-3 protein	Pores ion channels	mmu:118446
Thymidine kinase 2, mitochondrial	Nucleotido metabolism	---
Glyceraldehide 3-phosphate deshydrogenase	Glycolysis/Gluconeogenesis Carbon metabolism Biosynthesis of amino acids Hif-1 signaling pathway Alzheimer’s disease	mmu:14433
Alpha/beta hydrolase domain-containing protein 11	Serine peptidases	hsa:83451
Junctional adhesion molecule B	Cell adhesion molecules Tight junction Leukocyte transendothelial migration	mmu:67374
AMMECR1-like protein	Signaling proteins	mmu:225339
Protein N-terminal asparagine amidohydrolase	Ubiquitin system	mmu:18203
Mitochondrial fission factor	Mitochondrial biogenesis	mmu:75734
Major prion protein	Ferroptosis Prion disease—Mus musculus (house mouse) Pathways of neurodegeneration	mmu:19122
Peroxisomal membrane protein 11B	Peroxisome	mmu:18632
Lysophospholipid acyltransferase LPCAT4	Glycerophospholipid metabolism Ether lipid metabolism	mmu:99010
Eukaryotic translation initiation factor 4E-binding protein 1	Egfr tyrosine kinase inhibitor resistance Hif-1 signaling pathway Mtor signaling pathway Pi3k-Akt signaling pathway Longevity regulating pathway Insulin signaling pathway	mmu:13684

## Data Availability

Data is contained within the article.

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
