# Peer review of "Variation of the 2D Pattern of Brain Proteins in Mice Infected with Taenia crassiceps ORF Strain"

_ijms, 2024, doi:10.3390/ijms25031460_

Round 1
Reviewer 1 Report
Comments and Suggestions for Authors
This study uses 2D gels to identify proteins expressed in brain tissue of mice infected with .. The authors identified different patterns of protein expression following 2, 4 or 8 weeks post-infection; However a major limitation of these types of proteomic studies is that typically only proteins that are most abundant are identified. Proteins that are lowly expressed are often not detected. Thus, the approach does not account for all potentially relevant proteins. The authors need to discuss the limitations associated with their approach.
It would be most helpful if the list of proteins were represented as part of a functional group (i.e., signaling, cytoskeleton, etc…) in a separate table.
Given that intact brain tissue was used for these studies, there is no information regarding the spatial expression of proteins across brain regions or which cell types (endothelial, astrocytes, neurons…etc) might be expressing the proteins of interest. Again this is a limitation of the approach which should be discussed. The authors could refer to the literature and make some predictions about spatial and cell type expression of proteins.
Figure 1. The legend for figure 1 requires more information. For example, what do the red symbols (crosses) represent and how do they compare to the dark black spots? Define W2, W4 and W8.
Figure 2. The 2D gels seem to suggest that high-molecular weight proteins are not detected. This is a limitation of the approach and should be thoroughly discussed.
Author Response
January 11, 2024
Prof. Dr. Lei Zhou
Guest Editor
International Journal of Medical Sciences
Special Issue "Proteomics and Its Applications in Disease 2.0
This letter is in response to the reviewers' comments to our Manuscript ID: ijms-2802467 entitled “Variation of the 2D pattern of brain proteins in mice infected with Taenia crassiceps ORF Strain”.
Thank you very much for the valuable comments of the Reviewers. The Reviewers comments have been fully taken into account in the revised version of the manuscript. The changes are highlighted in yellow in the new text. None of the reviewers has suggested that the manuscript should undergo extensive English editing
REFEREE 1
This study uses 2D gels to identify proteins expressed in brain tissue of mice infected with .. The authors identified different patterns of protein expression following 2, 4 or 8 weeks post-infection; However a major limitation of these types of proteomic studies is that typically only proteins that are most abundant are identified. Proteins that are lowly expressed are often not detected. Thus, the approach does not account for all potentially relevant proteins. The authors need to discuss the limitations associated with their approach.
It would be most helpful if the list of proteins were represented as part of a functional group (i.e., signaling, cytoskeleton, etc…) in a separate table. Given that intact brain tissue was used for these studies, there is no information regarding the spatial expression of proteins across brain regions or which cell types (endothelial, astrocytes, neurons…etc) might be expressing the proteins of interest. Again this is a limitation of the approach which should be discussed. The authors could refer to the literature and make some predictions about spatial and cell type expression of proteins.
Figure 1. The legend for figure 1 requires more information. For example, what do the red symbols (crosses) represent and how do they compare to the dark black spots? Define W2, W4 and W8.
Figure 2. The 2D gels seem to suggest that high-molecular weight proteins are not detected. This
is a limitation of the approach and should be thoroughly discussed.
Indeed, low-expression proteins that could be relevant would not be considered in the study; however, despite this premise, I am not aware of a proteomic study where that situation has occurred. That is, while it is possible, it is unlikely. Nevertheless, the referee is correct in pointing out that this limitation should be included in the discussion section of the manuscript.
Lines have been added to the manuscript regarding this point, and they are found and highlighted in yellow (page 2).
A table has been added, including the proteins detected as part of a functional group. (Table 2)
The reasoning behind processing the entire brain instead of spatially expressing proteins was that, based on the results obtained, a group of proteins specifically associated with a region or a cell nucleus linked to a function could be identified. However, we did not find evidence of that. According to the results, the found proteins do not belong to a specific cerebral process but are ubiquitous proteins that support the idea of a generalized degenerative process in the brain rather than one confined to a particular region. Another possibility is that by processing the entire brain, the expression change occurring in a limited region of the brain may be "diluted." As suggested by the referee, this point is included in the discussion. (Page 10 first paragraph)
In Figure 1, it has been added that the red crosses represent the center of the spot. The legend of Figure 1 specifies that W2, W4, and W8 correspond to Weeks 2, 4, and 8, respectively.
Due to the sample processing for a 2D gel, the quaternary structure of proteins disappears, and only monomers, mostly of low molecular weight, are manifested. We do not believe that this aspect, which is purely technical, needs to be discussed in detail.
REFEREE 2
I In the present study, one would expect that the authors would advance their earlier observation by looking more closely into the host response in terms of cytokines, infiltrating immune cells and tissue damage in the infected brain. Instead they identify some protein changes by 2D gels by reference to other publications but they do not give any direct identification by western, immunocytochemistry or PCR. They vaguely mention that the proteins identified at 2,4,8 weeks post-infection correspond to different stages of the brain tissue response to the infection but no evidence is provided about crucial elements of this experimental design such as the clinical mouse state, pathology and immunopathology or the presence and load of parasite using microbiology or molecular tools. For these reasons, although interesting I consider the study very preliminary for publication before the execution of crucial experiments that compare protein changes to immune and the infection state and thus advance knowledge beyond the data presented in the earlier publication by Jimenez et al. in Pathogens. 2023
The referee mentions that he/she would expect an advance in the earlier observation by looking more closely into the host response in terms of cytokines, infiltrating immune cells, and tissue damage associated with the infection. Honestly, I don't understand why it should be so. Fortunately, scientific research is a path with multiple doors. Anyway, there are already several reports in the literature describing the action of cytokines and immune cells in brain damage in the context of a parasitic infection. Some of these reports are cited in the present study. In the literature review, we realized that there was no report addressing a study at the whole-brain level rather than by regions, on the differences in the expression of brain proteins in the infective process of T. crassiceps. That was the motivation for the present work.
The referee makes an assumption that we consider incorrect. The referee assumes that this is a work subsequent to the one mentioned in Pathogens 2023, and that is not the case. This work should be viewed from the perspective that it is complementary to the one in Pathogens 2023 and Acta Tropica 212:105696 2020, even though it may be published at a later date. It is a complementary work because the samples from this study were processed at the same time as the samples from the work in Pathogens 2023 and the work in Acta Tropica 212:105696 2020, essentially by the same authors. Of course, we understand that the referee could not have known this, but we believe that she/he can appreciate the budgetary, student, and time constraints that often influence the outcome of a publication.
We agree with the referee that the paper could improve in characterizing the described proteins. However, we believe that the isoelectric point and molecular weight coordinates, along with their alignment with previously identified brain proteins reported in the literature, lead us to make a deduction that is not insignificant and is worthy of consideration based on the presented data. The characterization of the proteins reported in the study, at the sequence and antibody levels, is in progress. In the manuscript, the Acta Tropica reference is added (Reference 49) and highlighted in yellow. Additionally, a paragraph is included in the discussion outlining the guidelines to be followed in terms of sequencing and strengthening the identification of spots with antibodies. (Page 11 first paragraph)
Sincerely
Dr. Pedro Ostoa Saloma

Reviewer 2 Report
Comments and Suggestions for Authors
In the present study, one would expect that the authors would advance their earlier observation by looking more closely into the host response in terms of cytokines, infiltrating immune cells and tissue damage in the infected brain. Instead they identify some protein changes by 2D gels by reference to other publications but they do not give any direct identification by western, immunocytochemistry or PCR. They vaguely mention that the proteins identified at 2,4,8 weeks post-infection correspond to different stages of the brain tissue response to the infection but no evidence is provided about crucial elements of this experimental design such as the clinical mouse state, pathology and immunopathology or the presence and load of parasite using microbiology or molecular tools. For these reasons, although interesting I consider the study very preliminary for publication before the execution of crucial experiments that compare protein changes to immune and the infection state and thus advance knowledge beyond the data presented in the earlier publication by Jimenez et al. in Pathogens. 2023
Comments on the Quality of English LanguageOne would expect that in this paper the authors would advance their earlier observation (Jimenez et al Pathogens 2023) by looking more closely into the host response in terms of cytokines, infiltrating immune cells and tissue damage in the infected brain. Instead the authors identify some protein changes by 2D gels by reference to other publications but they do not give any direct identification by western, immunocytochemistey or PCR. They vaguely mention that the proteins identified at 2,4,8 weeks post-infection correspond to different stages of the brain tissue response to the infection but no evidence is provided about crucial elements of this experimental design such as the clinical mouse state, pathology and immunopathology or the presence and load of parasite using microbiology or molecular tools. For these reasons, although interesting I consider the study very preliminary for publication. However, I encourage the authors to future resubmission after execution of crucial experiments that compare protein changes to immune and the infection state and thus advance knowledge beyond the data presented in the earlier publication by Jimenez et al. in Pathogens. 2023
Author Response

(The authors gave the same response as above.)

Round 2
Reviewer 1 Report
Comments and Suggestions for Authors
The authors have responded to the reviewer's comments/concerns and have made appropriate changes.